# The effects of mental fatigue on explicit and implicit contributions to visuomotor adaptation

David Apreutesei⬭, Erin K. Cressman⬭*

Faculty of Health Sciences, School of Human Kinetics, University of Ottawa, Ottawa, Ontario, Canada

* erin.cressman@uottawa.ca

**Data Availability Statement:** All relevant data are within the paper and its Supporting information files. Summary data files are available on the OSF repository at https://osf.io/wx35s/.

## Abstract

The goal of the current research was to establish the impact of mental fatigue on the contributions of explicit (i.e., conscious strategy) and implicit (unconscious) processes to visuomotor adaptation. Participants were divided into two groups, a Mental Fatigue (MF) group who completed a cognitively demanding 32-minute time load dual back task (TLDB), and a Control group who watched a documentary for a similar length of time. Following the TLDB task or documentary watching, participants trained to reach with a visuomotor distortion, such that cursor feedback was rotated 40° clockwise relative to hand motion. Explicit and implicit contributions to visuomotor adaptation were assessed following 3 blocks of 45 rotated reach training trials and again following a 20-minute rest. Levels of mental fatigue, as indicated on a self-report scale, increased significantly for the MF group following the TLDB task. The Control group did not display a similar increase in mental fatigue following the documentary watching. Results then revealed a decrease in visuomotor adaptation early in training for the MF group compared to the Control group, as well as decreased retention of visuomotor adaptation immediately following the 20-minute rest. Furthermore, correlational analyses revealed that greater levels of mental fatigue reported by participants in the MF group were associated with less explicit adaptation and greater implicit adaptation. Similar trends were not observed for the Control group. Taken together, the decreased visuomotor adaptation observed early in training, as well as the moderate correlation between increased mental fatigue and decreased explicit adaptation, suggest that mental fatigue decreases one's ability to engage in explicit processing, limiting the overall extent of initial visuomotor adaptation achieved.

## Introduction

Skill maintenance is the ability to maintain movement performance under changing conditions [1]. The current research examined the impact of mental fatigue on skill maintenance, or, as more commonly referred to in the literature, motor adaptation [2–4]. Our ability to adapt our movements to environmental changes is critical to ensure continued task success. Motor adaptation can be studied in a laboratory setting by having participants reach in a novel

**Funding:** This work was supported by a Discovery Grant provided by the Natural Sciences and Engineering Research Council of Canada (EKC; Grant Number: RGPIN-2018-04160). The funders had no role in this study. There was no additional external funding received for this study.

**Competing interests:** The authors have declared that no competing interests exist.

visual environment. For example, participants reach from a home position to a target with a robot manipulandum while the position of their hand is misrepresented on the screen by a cursor. Specifically, the cursor's trajectory can be rotated clockwise (CW) or counterclockwise (CCW) relative to a participant's actual hand motion [2,3,5–9]. When first reaching with the rotated cursor feedback, participants typically experience an error, as the cursor does not land on the target as expected. Participants then adjust their reaches over training trials to counteract the cursor rotation, demonstrating visuomotor adaptation [2,4,10–12]. Furthermore, when the cursor rotation is removed, participants display 'aftereffects', as they continue to reach as if the cursor rotation is still present [3,7,9,13,14].

The presence of aftereffects is proposed to reflect implicit (i.e., unconscious) visuomotor adaptation and arises due to participants experiencing a sensory prediction error (i.e., the sensory feedback arising from the movement does not match expected sensory feedback; [9,15]). Recent findings suggest that explicit processes (i.e., conscious strategy) are also engaged during training trials when reaching with a cursor rotation that is larger than 30˚ [16–18]. Explicit processes have been shown to contribute to visuomotor adaptation in the early stages of reach adaptation training, with their contribution decaying over trials. In contrast, implicit processes have been shown to arise later and contribute to a greater extent as training trials progress [4,12,17].

The contributions of explicit and implicit processes to visuomotor adaptation have been established using the Process Dissociation Procedure (PDP; [16]). Within the PDP of Werner and colleagues [16], participants are instructed to reach while (1) using what they learned during training trials (i.e., inclusion trials) and (2) refraining from using what they learned and to reach directly to the target (i.e., exclusion trials). Implicit adaptation is determined based on reaching errors on exclusion trials, while the difference in reaching errors between inclusion and exclusion trials is used to establish explicit adaptation.

To date, visuomotor adaptation and the processes underlying visuomotor adaptation have been studied extensively under 'ideal' conditions in a laboratory setting, which are not necessarily reflective of daily life. For example, previous investigations into visuomotor adaptation have not considered the potential impact of mental fatigue on performance, where mental fatigue is defined as "a psychobiological state caused by prolonged and/or intense periods of demanding cognitive activity and characterized by feelings of tiredness and lack of energy" [19]. Altered states of consciousness and reduced alertness permeate our day to day lives. In fact, according to an American survey conducted by the National Sleep Foundation in 2020, adults feel tired 3 days a week on average, with 28% of respondents stating that they feel tired 5–7 days a week [20].

The goal of the current experiment was to determine the impact of mental fatigue on explicit and implicit contributions to visuomotor adaptation. Previous research has shown that mental fatigue can influence both cognitive and motor performance. In particular, mental fatigue has been shown to compromise cognitive control processes [21,22], as demonstrated by working memory deficits [23,24], reductions in goal-directed attention [25–27] and response preparation [21,28,29], as well as impaired performance monitoring [22] and information processing [30]. More recent work has established that mental fatigue also influences motor control directly, with participants displaying increased movement time [19] and decreased performance on gross and fine motor tasks [31] following the completion of a mentally fatiguing task. Jacquet et al. [19] and Magnuson et al. [31] further showed that motor performance remains impaired relative to pre-fatigue levels when assessed at 15–20 minutes following completion of a mentally fatiguing task. In fact, Magnuson and colleagues [31] found that motor performance only recovered to baseline levels at 40 minutes following completion of the mentally fatiguing task, demonstrating a residual impact of mental fatigue on motor performance.

In the current study we asked if mental fatigue impacts visuomotor adaptation and underlying explicit and implicit contributions over time. Participants trained to reach with a large cursor rotation (i.e., 40˚), that has consistently been shown to engage both explicit and implicit processes [16,17]. Given that mental fatigue has been shown to impact cognitive performance and motor control (see [19,22,31]), it was hypothesized that the MF group would show decreased visuomotor adaptation compared to the Control group early in training, when explicit processes have been shown to be engaged [4,17]. Furthermore, it was hypothesized that explicit visuomotor adaptation as established via the PDP would be less in the MF group compared to the Control group over the course of the experiment. These results would indicate that mental fatigue impacts the conscious strategic component of visuomotor adaptation, while leaving the unconscious contribution largely unaffected.

## Methods

### Participants

An a priori power analysis using G*Power 3.1 with the following input parameters: a power of 0.8, expected effect size of 0.89 (based on the results of Neva et al. [32]) and a Type 1 error (alpha) probability of 0.05, revealed a total sample size of 12 participants was required. However, participant number was increased to 40 in order to match previous research [31]. Participants were recruited from the University of Ottawa and surrounding community to participate in this study and were randomly assigned to one of two groups; (1) Mental Fatigue (MF, n = 20, Mean age = 20.1 ± 1.8 years; 13 females) or (2) Control (n = 20, Mean age = 19.6 ± 1.6 years; 13 females). This experiment was approved by the University of Ottawa Health Sciences and Science Research Ethics Board. All participants provided written informed consent prior to testing and were informed that they could withdraw at any time.

All participants were naïve to the scope and purpose of the study and had never participated in a visuomotor adaptation experiment before. Participants were healthy, reporting no history of neurological disorders and/or sensorimotor dysfunction. As well, participants had normal or corrected-to-normal vision and all participants were right-handed (Edinburgh Handedness Questionnaire Mean Score = 85.5 ± 15.0; [33]).

Upon arriving at the laboratory, participants confirmed they had slept for at least 7 hours the previous night and had not consumed alcohol or engaged in strenuous physical activity the day (i.e., 24 hours) prior to the experiment. Furthermore, they indicated they had not consumed caffeine and/or nicotine in the 3 hours prior to the start of testing. In attempt to control for different levels of cognitive fatigue/effort present in each participant's day, data were collected before 1 pm.

### Experimental overview

The experiment was completed in a single testing session that lasted approximately two hours. In general, participants completed three tasks; (1) a mental fatigue scale (MFS; adapted from Lee et al. [34]), (2) a time load dual back task (TLDB; [35]) to induce mental fatigue or quiet rest, and (3) reaches to visual targets when visual feedback in the form of a cursor on the screen was aligned with participant hand motion, rotated relative to hand motion, or not present. See Fig 1 for an overview of the different tasks completed over the course of the experiment.

**Mental Fatigue Scale (MFS).** Participants completed the MFS a total of 7 times over the course of the experiment (see Fig 1). The MFS consisted of one question found in Lee et al.'s [34] Visual Analog Scale for Fatigue (VAS-F). Specifically, participants used a pen to annotate on a 10 cm line spanning from 0 to 100 their present feeling of mental fatigue, with 0 representing no mental fatigue at all and 100 representing extreme mental fatigue. After completing the MFS each time, the form was collected from the participant.

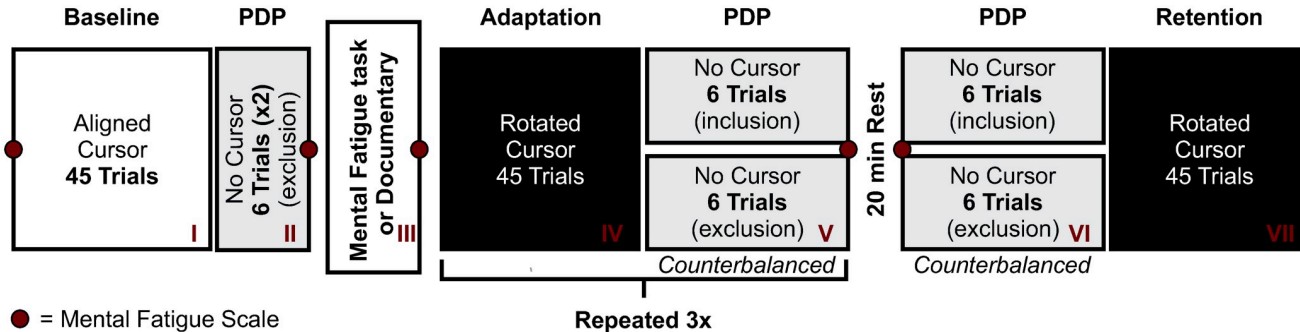

**Fig 1. Experimental procedure.** The experiment began with 45 aligned reach training trials (i.e., baseline), followed by 12 no cursor exclusion trials. This was followed by participants completing either a 32-minute mentally fatigue protocol (MF group) or watching a documentary (Control group). Following this, 45 rotated reach training trials were completed, in which participants reached with cursor feedback that was rotated by 40º relative to hand motion. These rotated reach training trials were followed by 6 no cursor exclusion and inclusion trials. The sequence of rotated reach training trials and no cursor reaches was completed 3 times. All participants then completed a 20-minute rest period. Finally, participants completed 6 no cursor exclusion and inclusion trials followed by 45 rotated reach training trials to assess retention. The mental fatigue scale (MFS) was administered to participants 7 times over the course of the experiment, as indicated by placement of the red dots, to track subjective feelings of mental fatigue throughout the experiment.

**TLDB task or rest.** Participants in the MF group completed the TLDB of [35]. This mental fatiguing task began with participants sitting in front of a computer screen (Dell P2210–22" Flat Panel Monitor; refresh rate = 60 Hz). Participants were informed of the two-component nature of the task. First, participants needed to pay attention to the white letters displayed on the black screen, which were from the following set of 8 letters: A, C, T, L, N, E, U and P. Letters were displayed in Arial font (size 16) in the middle of the screen for 1.5 seconds in duration. If the letter on the screen was the same as the previous letter shown, participants were to press the spacebar with their left index finger to indicate a match. Secondly, alternating with the letters shown on the screen, a number from a set of integers (1, 2, 3, 4, 6, 7, 8, 9) flashed on the screen for 1.5 seconds in duration. Participants were required to use their right index finger to press the number 2 on the keyboard if an even integer was displayed and the number 3 on the keyboard with their right middle finger if an odd integer was displayed. Letters and numbers alternately flashed on the screen for a total task time of 32 minutes (i.e., 1260 items were displayed) and participants were to make their responses as fast and accurately as possible. During this time the Control group watched a documentary (i.e., 32 minutes; Animal by BBC, Season 1 ep. 1: Big Cats; Netflix, Inc.).

### Reaching task—Visuomotor adaptation

**Apparatus.** Testing took place using the KINARM End-Point Lab (KINARM Technologies, Kingston, ON). As shown in Fig 2, visual targets were projected from a downward facing monitor (LG 47LD452B-UA EzSign– 47" LCD TV; refresh rate = 60 Hz), located 20.5 cm above a reflective surface that was located 20.5 cm above the robot handle, ensuring that visual stimuli appeared to lie in the same horizontal plane as the right hand holding the robot handle. Participants seated themselves in a height and tilt adjustable chair located in front of the experimental apparatus. Participants adjusted the position of the chair so that their forehead rested against the testing apparatus and they could reach comfortably to all targets presented within the workspace. Once comfortable, the position of the chair was maintained throughout all reaching trials. Participants' view of their limbs was occluded by the reflective surface and a black cloth that was draped around their neck and attached to the apparatus. The room lights were also turned off.

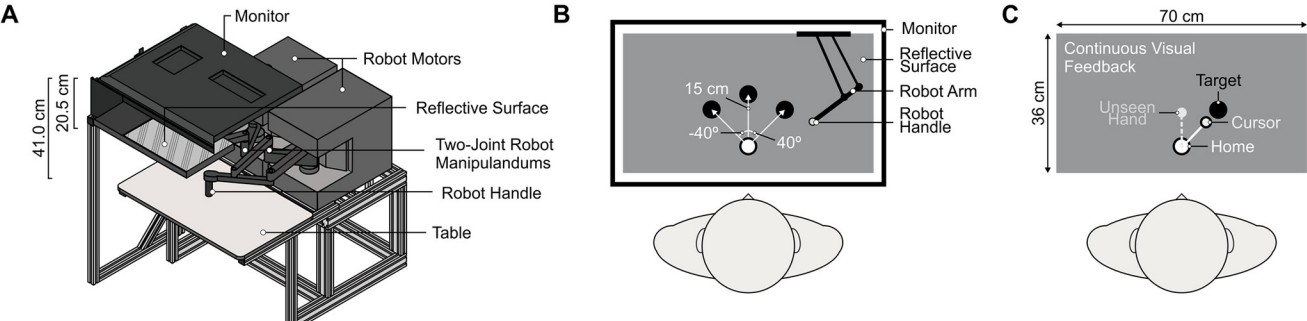

**Fig 2. Experimental apparatus and visual display. (A)** View of the KINARM experimental apparatus. Participants grasped the robot manipulandum with their right hand. While reaching, participants did not have direct vision of their hand, but could see visual (i.e., cursor) feedback regarding the position of their hand. **(B)** Aerial view of the 3 target positions participants reached to during the reach training trials. The first target was 15 centimeters in front of the home position, while the other two targets were 15 cm from the home position at 40° to the right and left of center. **(C)** Example of a reach training trial in which cursor feedback was rotated 40° clockwise relative to hand motion. Participants needed to reach 40° to the left of the target to compensate for the rotated cursor feedback and hence have the cursor land on the target.

**Reach training trials.** Participants grasped the robot handle with their right hand. An aligned (baseline) reach training trial began with the participant's hand held in the home position for 500 ms, where the home position (white circle, 2 cm in diameter) was located approximately 20 cm in front of their chest, in line with their body midline (Fig 2B). After 500 ms, one of three targets appeared (yellow circle, 2 cm in diameter) at a distance of 15 cm from the home position. The targets appeared straight ahead of the home position (central 0° target) or 40° to the left or right of center. Participants were instructed to reach to the target with the goal of having the cursor land on the target. Real-time visual feedback of the hand position was provided by the cursor on the screen (magenta circle, 1 cm in diameter) throughout the movement. Once the cursor landed on the target, (i.e., the center of the cursor and the center of the target were within 0.5 cm), the hand was held at this position for another 500 ms. The cursor and target then disappeared. Finally, the robot passively moved the hand back to the home position following a direct, linear path in a movement time of 1000 ms. If participants attempted to move outside of the linear path, a resistance force (proportional to the depth of penetration with a stiffness of 2 N/mm and a viscous dampening of 5 N/mm) perpendicular to the grooved path was produced. The position of the KINARM robot was recorded at a sampling rate of 1000 Hz, with a spatial accuracy of 0.1 mm. Participants completed 45 aligned reach training trials at the beginning of the experiment. See Fig 3 for a timeline of events for aligned reach training trials.

The rotated reach training trials proceeded in a similar manner. However, this time, the cursor feedback representing the position of the hand was rotated 40° clockwise (CW) relative to the participant's actual hand trajectory. Participants were not made aware of the rotation ahead of time, nor given instructions on how to counteract the visuomotor rotation, and no questions regarding the rotation were answered. As shown in Fig 1, participants completed 4 blocks of 45 rotated reach training trials over the course of the experiment to assess the impact of mental fatigue on visuomotor adaptation over time (i.e., early vs. late visuomotor adaptation), and retention of visuomotor adaptation.

**No cursor reaches.** In these reaches, participants reached when no cursor feedback was displayed. Following aligned and rotated reach training, participants performed no cursor reaches under exclusion instructions in which they were instructed:

*"You are now going to reach when you cannot see your hand, as there will be no cursor on the screen. For these trials, do not use anything you may have learned to get the cursor to the*

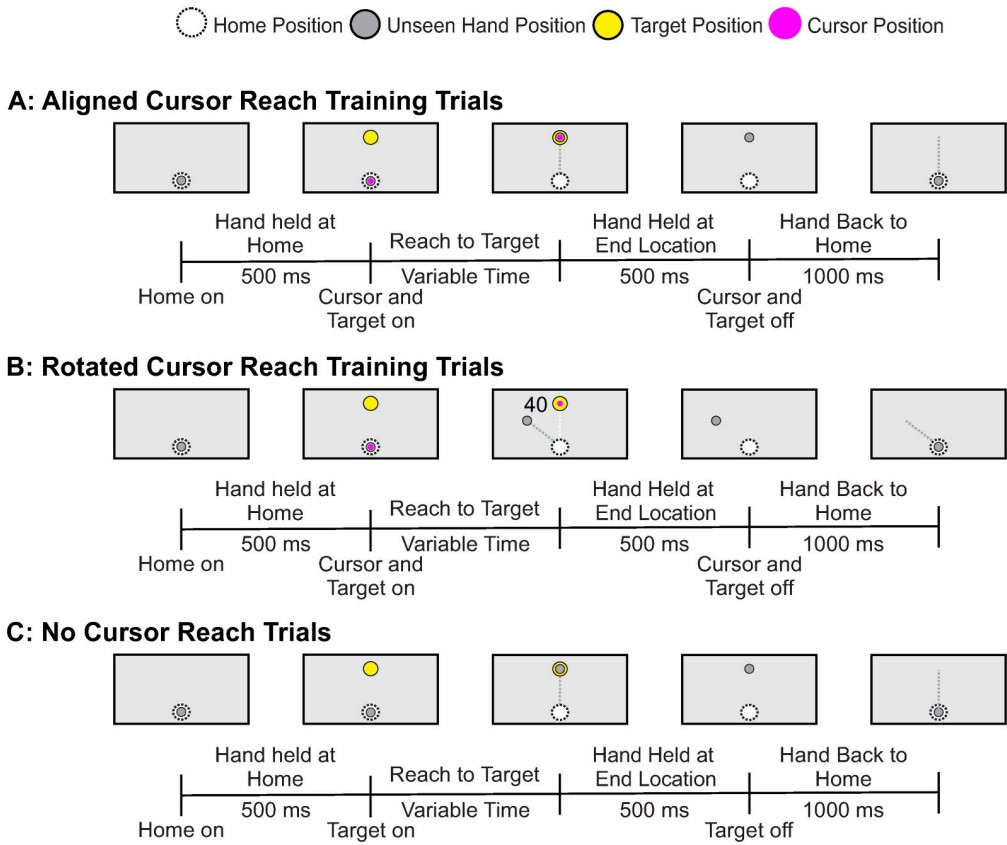

**Fig 3. Timeline of reaching trials. (A)** Aligned reach training trials. Participants reached to a target with aligned cursor feedback. **(B)** Rotated reach training trials. Participants reached to a target with rotated cursor feedback, in which the cursor's trajectory was rotated 40° clockwise relative to hand motion. **(C)** No cursor reach trials. Participants reached to a target without receiving any visual feedback regarding their hand's position in space.

*target. Instead, aim so that your hand goes straight to the target as you did during the baseline reaches."*

Participants performed 12 exclusion trials following the block of aligned reach training (4 trials to each target) and 6 exclusion trials following each block of rotated reach training (2 trials to each target).

When completing the block of no cursor reaches following rotated reach training, participants also performed 6 reaches under the inclusion instruction:

*"You are now going to reach when you cannot see your hand, as there will be no cursor on the screen. For these trials, use anything you have learned during training in order to get the cursor to the target. In other words, aim so that the cursor would have gone straight to the target, as in the training trials you just completed."*

During the no cursor trials, participants reached to the target with no visual feedback and held the handle at the location they assumed to best correspond with the target. Once the velocity of their movement dropped below 0.01 m/s for more than 100 ms, the reach was considered complete. The hand was then held at the final position for 500 ms before the robot

handle brought the hand back to the home position. The order in which inclusion and exclusion trials were completed was counterbalanced across participants.

## Data analyses

Following confirmation that data met the required assumptions for the statistical test used, statistical analyses were completed using parametric statistics in IBM SPSS Statistics for Windows version 28.0 unless otherwise noted. The Greenhouse-Geisser correction was applied when appropriate and corresponding p-values are reported. For data that were not normally distributed according to the Kolmogorov-Smirnov test (see Final adaptation: Late training), the non-parametric Mann-Whitney *U* test was used to test for group differences. The significance value for all statistical tests was set at *p < 0.05* and Bonferonni post-hoc tests corrected for multiple comparisons were used to find the locus of significant effects or interactions when required.

### MFS: Assessment of fatigue

Responses on the MFS were determined for each participant for each of the 7 times it was completed. A response was the measured value on the continuous line in millimeters between 0 (no mental fatigue) to 100 (extreme mental fatigue) that corresponded to the location at which participants indicated their current level of mental fatigue. The first MSF response for each participant (i.e., Time 1) was considered baseline performance and used to normalize all other responses. The remaining 6 normalized responses were then compared across groups in a 2 Group (MF vs. Control) x 6 Time mixed analysis of variance (ANOVA) with repeated measures (RM) on the last factor to determine changes in mental fatigue over the course of the experiment. We looked to establish that mental fatigue increased for the MF group following the TLDB task (i.e., Time 3) relative to levels reported following the completion of the aligned training block (i.e., Time 2), and confirm that the Control group did not show a similar increase following the documentary watching. In the S1 File we report findings of our analyses related to Times 4 through 7, which provide insight into whether completing visuomotor adaptation trials led to increased mental fatigue.

### TLDB: Induction of fatigue

Initial and final accuracy, as well as mean reaction time of responses on the TLDB task for the MF group were compared in paired samples t-tests to ensure that participants remained engaged in the task over the 32 minutes as seen by Jacquet and colleagues (i.e., performance levels were maintained; [19]). Initial and final accuracy were taken as the percentage of correct responses over the first or last 315 trials completed in the TLDB task respectively (i.e., the first or last 25% of the total number of trials completed). These percentage data were arcsine transformed before undergoing analysis. Initial and final mean reaction times were calculated to be the mean reaction time across the first or last 315 trials completed, respectively. Details regarding additional analyses and results related to stimulus type over the entire TLDB task can be found in the S1 File.

### Reaching trials: Visuomotor adaptation

All reaching trials (i.e., aligned reach training trials, rotated reach training trials and no cursor reaches) were analyzed using a custom written MATLAB program. Within the MATLAB program, the resultant velocity in the x-y plane was determined at each millisecond within a trial using the x-y positional coordinates recorded, where a trial was defined from home position

onset until the robot began to move the handle back to the home position. The resultant velocity of each trial was then plotted over time and used to establish movement onset and movement end. Movement onset and movement end were determined by selecting the time points at which resultant velocity increased above or decreased below 0.01 m/s and was maintained above or below 0.01 m/s for 100 ms, respectively. The time corresponding to peak velocity within the movement was then found and the corresponding x-y positional coordinates used to calculate the angular error at peak velocity (PVAE), where PVAE is the difference in degrees between a vector connecting the home position to the desired target and a vector connecting the home position to the position of the hand at resultant peak velocity.

PVAE was used to screen for outliers, with trials considered outliers if their absolute PVAE was more than 3 standard deviations above the mean PVAE of the corresponding block of trials. In total, 84 trials were initially identified as outliers. However, 32 of these trials were the first trial found within a rotated training block and showed similar PVAE to the aligned reach training trials, as expected. Thus, these trials were kept in the analysis, leading to the removal of 52 trials in total (or 0.46% of all trials).

**Initial adaptation: Early training.** The mean PVAE of the last 10 aligned reach training trials were compared between the MF and Control groups using an independent samples t-test to check for differences between groups with respect to baseline reaching performance. We then determined the impact of inducing mental fatigue on initial reach adaptation by comparing the mean PVAE of the first 10 rotated reach training trials across the 4 rotated training blocks between the MF and Control groups. These means were normalized by subtracting the mean PVAE of the last 10 aligned reach training trials. The normalized mean PVAE were compared across groups in a 2 Group (MF vs. Control) x 4 Block (1st, 2nd, 3rd adaptation vs. retention block) mixed ANOVA with RM on the last factor.

**Final adaptation: Late training.** Final levels of reach adaptation were compared across groups to establish if mental fatigue had an impact on the extent of visuomotor adaptation achieved. To do this, the mean PVAE was found for the last 10 trials of the 4 rotated training blocks. These values were again normalized relative to the mean PVAE of the last 10 aligned reach training trials. For each Block (1st, 2nd, 3rd adaptation and retention block), normalized values were compared across Groups using a Mann-Whitney $U$ test.

**No cursor reaches.** The no cursor trials were analyzed to directly compare the contributions of explicit and implicit adaptation to visuomotor adaptation between the MT and Control groups. The mean PVAE of the no cursor reaches following the aligned cursor block were compared between the MF and Control groups in a 2 Group (MF vs. Control) x Set of 6 trials (1st vs. 2nd set of 6 trials) mixed ANOVA with RM on the last factor to establish any differences in groups with respect to baseline reaching errors.

The implicit index (II) was then determined for each set of exclusion trials completed following rotated reach training trials (blocks V and VI, Fig 1) according to the following formula:

$$\text{Implicit index(II)} = M_{\text{PVAE Exclusion trials}}$$

The explicit index (EI) was calculated from the same blocks of trials following rotated reach training trials (blocks V and VI, Fig 1) according to the following formula:

$$\text{Explicit index(EI)} = M_{\text{PVAE Inclusion trials}} - \text{II}$$

Where $M_{\text{PVAE}}$ is the mean PVAE of the 6 exclusion or 6 inclusion trials that a participant completed. Each index was normalized by subtracting the $M_{\text{PVAE}}$ of the corresponding exclusion trials completed following aligned reach training (block II, Fig 1) to establish explicit and implicit adaptation. Specifically, mean PVAE was calculated for the first and second set of 6

exclusion trials following aligned reach training. If participants completed exclusion (or inclusion) trials immediately following rotated reach training, then the mean PVAE of the first 6 exclusion trials following aligned reach training was subtracted from the II (or EI). If participants completed the exclusion (or inclusion) trials after the inclusion (or exclusion) trials following rotated reach training, then the mean PVAE of the last 6 exclusion trials following aligned reach training was subtracted from the II (or EI). These calculations led to explicit or implicit adaptation being in the incorrect direction for 16 of the 320 values calculated (i.e., explicit or implicit adaptation was to the right of the target). These 16 values were removed from analyses and values imputed using the linear interpolation function in IBM SPSS Statistics for Windows version 28.0. Explicit and implicit adaptation were then compared between groups in a 2 Group (MF vs. Control) x 4 Block (1st, 2nd, 3rd rotation vs. retention block) mixed ANOVA with RM on the second factor to establish the impact of inducing mental fatigue on explicit and implicit contributions to visuomotor adaptation over time.

Finally, given the variance we observed across participants in the MF group with respect to changes in level of mental fatigue following the TLDB task and our hypothesis that increased mental fatigue would be associated with decreased explicit adaptation, one-tailed Pearson correlation analyses were completed to assess the relationship between both explicit and implicit adaptation and mental fatigue for the MF and Control groups. A participant's explicit and implicit adaptation were defined as their mean explicit and implicit adaptation across the 4 blocks of no cursor reaches following rotated reach training, while changes in mental fatigue were taken to be the difference between a participant's mental fatigue response immediately post TLDB task (MF group) or documentary watching (Control group; Time 3) and initial mental fatigue report (Time 1). In total, 4 correlations were performed, as explicit adaptation and implicit adaptation were correlated to changes in mental fatigue for both the MF and Control groups to establish the relationship between level of mental fatigue and explicit and implicit adaptation.

## Results

### Mental fatigue

Reports of mental fatigue over the course of the experiment are displayed in Fig 4A. ANOVA revealed a non-significant main effect of Group, suggesting that both groups reported similar mean mental fatigue across the experiment (MF: $\bar{X}$ = 17.0, SE = 3.3; Control: $\bar{X}$ = 12.2, SE = 3.2; $F(1, 38) = 1.093$, $p = 0.302$). However, ANOVA did reveal that mental fatigue differed significantly over Time ($F(3.0, 112.6) = 26.223$, $p < 0.001$), and there was a significant Group x Time interaction ($F(3.0, 112.6) = 6.318$, $p < 0.001$). Post-hoc analysis revealed that mental fatigue reported after the aligned training block, prior to the TLDB task or documentary watching (i.e., Time 2), was on average 4.1 (SE = 2.9) for the MF group and 0.4 (SE = 1.9) for the Control group, which did not differ significantly from each other ($p = 0.291$). Mental fatigue for the MF group then increased significantly immediately following completion of the TLDB task (Time 3), with mental fatigue being 25.4 points higher relative to mental fatigue reported prior to the TLDB task (Time 2; $p < 0.001$). This increase in mental fatigue for the MF group was greater than that reported by the Control group (Time 3; MF: $\bar{X}$ = 29.5, SE = 5.0; Control: $\bar{X}$ = 11.3, SE = 3.2; $p = 0.004$). Mental fatigue did not increase significantly in the Control group from Time 2 to Time 3 ($p = 0.145$).

### Time load dual back task

Participants in the MF group retained a high level of response across the TLDB task (Time Interval 1: $\bar{X}$ accuracy = 86.9%, SE = 2.5%; Time Interval 4: $\bar{X}$ accuracy = 91.2%, SE = 1.6%),

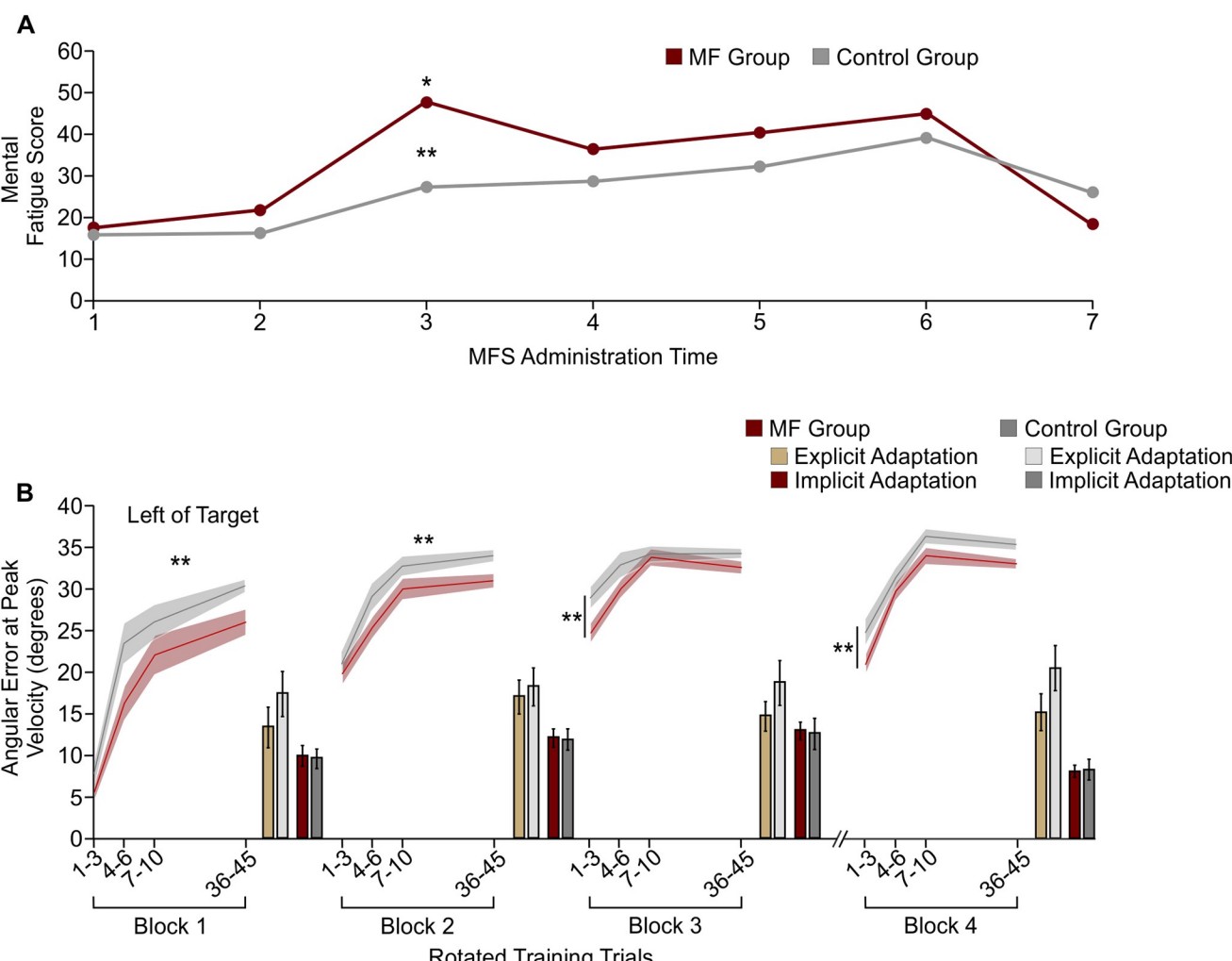

**Fig 4. Mental fatigue and reaching performance across the experiment.** (A) Mental fatigue scores for the MF group (burgundy) and Control group (grey) reported across the experiment. (B) Angular errors at peak velocity across blocks of rotated reach training trials. Data displayed are the average of 3 or 4 training trials early in each block (trials 1–10), or the last 10 training trials at the end of each block. Explicit and implicit adaptation are presented as bars following each rotated training block. Explicit and implicit contributions for the MF group are represented by gold and garnet bars respectively, whereas for the Control group they are represented by light grey and grey bars respectively. Shaded regions and error bars represent standard error of the mean. Asterisks (*) represent significant differences relative to levels of mental fatigue reported prior to completing the TLDB task or documentary watching (i.e., Time 2 vs. Time 3; *p < 0.05*). Double asterisks (**) represent significant differences between groups (*p < 0.05*).

and even demonstrated improved performance from initial to final task performance (t(19) = -3.070, *p = 0.006*). Reaction time results indicated that this improvement in accuracy did not arise due to slower responses, as there was a trend for reaction time to decrease across the TLDB task (Time Interval 1: $\bar{X}$ reaction time = 1.01 sec, SE = 0.01; Time Interval 4: $\bar{X}$ reaction time = 0.99 sec, SE = 0.01; t(19) = 2.019, *p = 0.058*).

## Visuomotor adaptation

**Early visuomotor adaptation.** Participants reached fairly accurately to the targets at the end of the aligned block of reach training trials with minimal errors (MF: $\bar{X}$ = 2.33˚, SE = 0.31˚; Control: $\bar{X}$ = 1.95˚, SE = 0.41˚). Independent samples t-tests confirmed that these final reaching errors in the aligned block did not differ between the MF and Control groups

(t(38) = 0.727, $p = 0.471$). As can be seen in Fig 4, reaching direction changed with the introduction of the visuomotor distortion, with participants began to reach to the left of the target.

Visuomotor adaptation differed between Groups during initial rotated reach training trials across the 4 blocks, as indicated by a significant main effect of Group (F(1,38) = 4.817, $p = 0.034$). Overall, participants in the MF group demonstrated less initial visuomotor adaptation than participants in the Control group across all 4 blocks of rotated training (MF: $\bar{X} = 25.56°$, SE = 0.76°; Control: $\bar{X} = 28.58°$, SE = 1.15°). Analyses also revealed a significant main effect of Block (F(2.3, 85.5) = 128.535, $p < 0.001$), but no significant Block x Group interaction (F(2.3, 85.5) = 0.885, $p = 0.427$). Post-hoc analysis confirmed that early visuomotor adaptation increased across the first three Blocks, with PVAE significantly greater in Block 2 compared to Block 1 ($p < 0.001$) and Block 3 compared to Block 2 ($p < 0.001$). There was no difference in early PVAE between Block 3 and Block 4, following the 20-minute rest ($p = 1$).

**Late visuomotor adaptation.** With respect to final levels of visuomotor adaptation achieved within each block, analyses revealed a significant difference between Groups following the first (*U*(N MF group = 20, N Control group = 20) = 301.00, z = 2.732, $p = 0.006$) and second (*U* (N MF group = 20, N Control group = 20) = 283.00, z = 2.245, $p = 0.024$) rotated training blocks. Specifically, the Control group demonstrated greater visuomotor adaptation than the MF group at the end of the first two blocks of rotated training (MF: rotated block 1$\bar{X} = 26.13°$, SE = 1.29°, rotated block 2$\bar{X} = 31.29°$, SE = 0.92°; Control: rotated block 1$\bar{X} = 30.52°$, SE = 1.14°, rotated block 2$\bar{X} = 34.43°$, SE = 0.98°). No significant differences were found between groups following the third (*U* (N MF group = 20, N Control group = 20) = 242.00, z = 1.136, $p = 0.265$) or fourth (*U* (N MF group = 20, N Control group = 20) = 260.00, z = 1.623, $p = 0.108$) rotated training blocks.

## No cursor reaches: Explicit and implicit visuomotor adaptation

Participants reached with no cursor feedback following aligned reach training. Again in these trials participants reached to the target across both sets of 6 trials with minimal mean PVAE (MF: $\bar{X} = 2.90°$, SE = 0.55°; Control: $\bar{X} = 2.81°$, SE = 0.66°; Group: F(1, 38) = 0.010, $p = 0.922$). Furthermore, reaching errors did not differ across the two sets of 6 trials (Set 1: $\bar{X} = 2.72°$, SE = 0.51°; Set 2: $\bar{X} = 2.98°$, SE = 0.45°; F(1, 38) = 0.316, $p = 0.577$).

**Explicit adaptation.** Explicit adaptation following each block of rotated training are displayed in Fig 4B. From this figure it appears that the MF group demonstrated less explicit adaptation compared to the Control group (MF: $\bar{X} = 15.14°$, SE = 1.76°; Control: $\bar{X} = 18.79°$, SE = 2.38°), however analyses indicated a non-significant main effect of Group (F(1, 38) = 1.523, $p = 0.225$), and a non-significant Block x Group interaction (F(3, 114) = 0.847, $p = 0.471$). As well, the main effect of Block was not significant (F(3, 114) = 1.390, $p = 0.250$). Thus, results suggest that explicit adaptation remained relatively constant across rotated training trials and did not differ significantly between groups.

**Implicit adaptation.** Similarly, results indicated that implicit adaptation did not differ between Groups (MF: $\bar{X} = 10.85°$, SE = 0.79°; Control: $\bar{X} = 10.63°$, SE = 1.20°; F(1, 38) = 0.022; $p = 0.883$). However, analyses did reveal a significant main effect of Block (F(2.3, 88.5) = 15.877, $p < 0.001$), but not a significant Block x Group interaction (F(2.3, 88.5) = 0.073, $p = 0.950$). Post-hoc analysis revealed that implicit adaptation increased from Block 1 ($\bar{X} = 9.89°$, SE = 0.77°) to Block 2 ($\bar{X} = 12.04°$, SE = 0.77°; $p = 0.006$), where it plateaued (i.e., there was no significant increase from Block 2 to Block 3: $\bar{X} = 12.85°$, SE = 1.07; $p = 1$). Implicit adaptation then decreased after the 20-minute rest in Block 4 ($\bar{X} = 8.18°$, SE = 0.72°), so that it was significantly lower than implicit adaptation in Block 3 ($p < 0.001$).

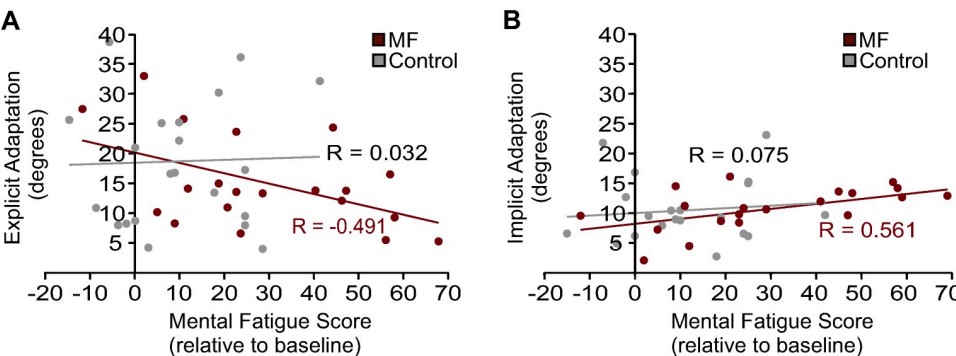

**Fig 5. Correlational analyses of explicit and implicit adaptation relative to mental fatigue.** Pearson correlations between (**A**) mean explicit adaptation, and (**B**) mean implicit adaptation relative to mental fatigue scores following the TLDB or documentary watching normalized to initial mental fatigue. Garnet circles represent data for participants in the MF group and grey circles represent data for participants in the Control group.

### Relationship of mental fatigue to explicit and implicit adaptation

In Fig 5A and 5B we show individual participant data, with each participant's mean (A) explicit and (B) implicit adaptation plotted relative to their change in mental fatigue scores after the TLDB (MF group) or the documentary viewing (Control group). As shown in Fig 5, changes in mental fatigue for the Control group were minimal, with scores on average only 11.3 points greater than baseline levels. Furthermore, participant data for the Control group are clustered on the left-hand side of Fig 5, demonstrating a consistency in perceived changes in mental fatigue reported by participants. In contrast, there was much greater variation in changes in mental fatigue relative to baseline levels for participants in the MF group, with changes in mental fatigue ranging from -12 to 69. The standard deviation for changes in mental fatigue for the MF group was 22.3 compared to 14.3 for the Control group.

Within the MF group a moderate negative correlation was found between changes in mental fatigue and mean explicit adaptation ($r(18) = -0.491$, $p = 0.014$), revealing that explicit adaptation was less for participants who reported higher levels of mental fatigue. Furthermore, a moderate positive correlation was found between changes in mental fatigue and mean implicit adaptation for the MF group ($r(18) = 0.561$, $p = 0.005$), indicating that implicit adaptation was greater for participants who reported increased mental fatigue. Analyses revealed that the correlation coefficients were weak for the Control group with respect to explicit adaptation ($r(18) = 0.032$, $p = 0.447$) and implicit adaptation ($r(18) = 0.075$, $p = 0.377$).

### Discussion

To date, the impact of mental fatigue on motor control, specifically visuomotor adaptation, has largely been left unexplored. In the current research we looked to establish the impact of mental fatigue on explicit (conscious strategy) and implicit (unconscious) processes underlying visuomotor adaptation. Mental fatigue was induced in one group of participants (i.e., the MF group) through a time load dual back (TLDB) task. Participants maintained a high level of accuracy over the course of the TLDB task (90% response accuracy overall), and responses were initiated quickly. Responses by participants in the MF group indicated that their mental fatigue had increased significantly at the end of the TLDB task. No such increase in mental fatigue was observed for the Control group, who sat and watched an animal documentary for a similar length of time. The MF group then exhibited decreased visuomotor adaptation early in

training (i.e., until the end of the third block of rotated training) compared to the Control group. While analyses revealed no difference in the extent of explicit or implicit adaptation across groups over the course of the experiment, we did find significant correlations between changes in mental fatigue and the extent of both explicit and implicit adaptation for participants in the MF group. Increased mental fatigue reported following the TLDB task was moderately correlated with explicit adaptation, with greater increases in mental fatigue reported by participants associated with less explicit adaptation. In contrast, a moderate positive correlation was found between increases in mental fatigue and implicit adaptation, with greater increases in mental fatigue associated with increased implicit adaptation. Explicit and implicit adaptation were weakly correlated with the minimal changes in mental fatigue reported by the Control group following watching of a documentary. Thus, results indicate that increases in mental fatigue impact visuomotor adaptation. In particular, we suggest that the ability of participants to engage in explicit adaptation is decreased with increased mental fatigue.

## Mental fatigue

For this research, we defined mental fatigue as a psychobiological state characterized by feelings of tiredness and lack of energy arising due to either prolonged and/or intense periods of demanding cognitive activity [19]. We had participants complete the TLDB task of [35] in hopes of inducing mental fatigue quickly. The TLDB task engages participants' short-term memory recall, as participants must remember letters shown on the screen for 1.5 seconds and respond if the current letter displayed corresponds to the previous one presented. While remembering letters, participants must also complete a choice reaction time task, in which they respond to even and odd numbers appearing on the screen as quickly as possible. The continuous and interchanging nature of the TLDB task requires constant attentional focus, which is presumed to drain limited attentional resources in accordance with the resource depletion theory [36]. The benefit of using the TLDB task is that it has reliably been shown to induce mental fatigue within a short timespan. For example, Borragan et al. [35] reported significant increases in mental fatigue following just 16 minutes of task completion, while Jacquet et al. [19] reported increased mental fatigue in their participants and impaired motor performance after completing the TLDB task for 32 minutes.

Previous research employing the TLDB has shown changes in neural activation associated with increased mental fatigue. For example, electroencephalography (EEG) recordings have indicated an increase in alpha band oscillations within the cerebral cortex over the course of completing the TLDB task for 32 minutes [19], a marker which has been accepted as reflecting decreased arousal and alertness [25,37,38], and thus considered as an indication of mental fatigue. These neural changes were found in parallel with increases in mental fatigue reported by participants on the same single question adopted from the Visual Analog Scale for Fatigue (VAS-F; [34]) that we used in the current experiment, promoting the use of the TLDB as a tool to increase mental fatigue and the MFS as a method to assess mental fatigue.

In accordance with [19,35], we saw increases in mental fatigue following completion of the TLDB task. Participants in our MF group indicated a significant increase in mental fatigue of 29.5 points following the TLDB task. This increase in mental fatigue was maintained above initial baseline levels until the end of the third rotated training block, approximately 20 minutes post TLDB task completion. Mental fatigue was only found to decrease to initial baseline levels following an additional 20-minute rest interval, or approximately 40 minutes after completion of the TLDB task. This lasting mental fatigue observed in our MF participants is in accordance with findings of Jacquet et al. [19], who found that mental fatigue following their TLDB task remained elevated above baseline levels for 20 minutes.

The lingering mental fatigue we observed throughout completion of our visuomotor adaptation trials could be due to participants completing the TLDB task. Alternatively, engagement in the visuomotor adaptation trials themselves could have led to the persistence of previously induced mental fatigue or even led to the introduction of additional mental fatigue. As discussed below, we found that participants engaged in explicit visuomotor adaptation (i.e., cognitive strategies). This explicit engagement may have increased mental fatigue over the course of the rotated training blocks. In support of this proposal, we saw an increase in mental fatigue over the course of the rotated training blocks for the Control group (as shown in the S1 File), with their reports of mental fatigue significantly greater than initial baseline levels by the end of the third rotated training block, even though they did not complete the TLDB task.

The increase in mental fatigue reported by our Control group over the course of the rotated reach training trials suggests that completion of a visuomotor adaptation task itself may increase mental fatigue. Future work is required to establish the relationship between mental fatigue arising due to completing a visuomotor paradigm and the impact of this mental fatigue on visuomotor adaptation. For now, based on our current results, we are confident that mental fatigue increased in the MF group following the TLDB task.

## Visuomotor adaptation: Explicit engagement

In general, visuomotor adaptation to a large cursor rotation (i.e., greater than 30˚) has been shown to involve both explicit and implicit processes [12,16,17,39]. These processes are proposed to be independent [12,39,40], with explicit processes contributing to a larger extent early in visuomotor adaptation [4], and implicit processes increasing their contribution as visuomotor adaptation progresses [4,41]. We hypothesized that mental fatigue would lead to decreased explicit adaptation given the decrease in cognitive resources available.

**Visuomotor adaptation trials.** As shown in Fig 4, visuomotor adaptation for both our MF and Control groups followed a logarithmic learning curve, with the greatest changes in reaches occurring within the early trials of the first rotated training block. The patterns of visuomotor adaptation we observed are similar to Heirani Moghaddam et al. [41], who used a similar experimental paradigm with respect to the initial number of blocks and trials completed (3 blocks of 45 rotated reach training trials). In agreement with Heirani Moghaddam et al. [41], we observed an increase in visuomotor adaptation within each block of trials and across blocks, such that adaptation was greatest by the end of the third block of rotated reach training for both our MF and Control groups.

While visuomotor adaptation patterns were similar between our two groups, the MF group adapted their reaches to a lower extent overall in early rotated reach training trials compared to the Control group. In fact, it was only by the end of the third rotated training block that reaching errors were similar across groups. Thus, mental fatigue led to a decreased level of reach adaptation at the start of each rotated training block and throughout the first two rotated training blocks, when explicit processes were expected to have a greater contribution [4,17].

We are aware of only one prior study (see [42]) that has looked to manipulate cognitive resources prior to participants completing a visuomotor adaptation task. In Anguera et al.'s [42] paradigm, participants completed a task designed to deplete spatial working memory resources prior to visuomotor adaptation trials. The spatial working memory depletion task required participants to mentally rotate a previously presented target shape and then indicate if it matched a subsequently displayed shape 3000 ms later. Results indicated that participants performed worse on a spatial working memory task immediately following the spatial working memory depletion task. Moreover, participants who performed worse on the spatial working

memory task also adapted their reaches to a lesser extent, leading to Anguera et al. [42] to conclude that working memory resource depletion negatively impacts visuomotor adaptation.

In agreement with Anguera and colleagues [42], we show that requiring participants to complete a mentally fatiguing task prior to visuomotor adaptation impairs the extent to which participants initially adapt their reaches. However, our current results call into question if the findings of Anguera et al. [42] arose due to the depletion of spatial working memory resources per se or because their participants experienced mental fatigue, which Anguera et al. [42] did not assess. In the current work, we show that the same decrements in early visuomotor adaptation found by Anguera et al. [42] arise with increased mental fatigue.

**No cursor reaches: PDP trials.**   We used the Process Dissociation Procedure (PDP; [16]) to assess explicit contributions to visuomotor adaptation directly. In contrast to our hypothesis, we found no significant group differences with respect to explicit (or implicit) visuomotor adaptation between groups. Explicit adaptation was 13.45° for the MF group and 17.48° for the Control group following the first block of rotated training trials and was maintained at a similar extent across the experiment. Implicit adaptation increased across initial rotated training blocks as seen by Heirani Moghaddam et al. [41].

While explicit adaptation did not differ significantly between groups overall, we found that mean explicit adaptation was approximately 4° less in the MF group compared to the Control group. Moreover, as shown in Fig 5, changes in mental fatigue following the TLDB task varied greatly among MF participants. A few participants reported no change in mental fatigue, while others reported a change in mental fatigue of up to 69 points. Taking individual changes in mental fatigue into account, analyses revealed that increased mental fatigue following the TLDB task was moderately (significantly) correlated with lower explicit adaptation in the MF group.

Implicit adaptation was also found to be associated with changes in mental fatigue in the MF group, but in the opposite direction, as increased mental fatigue was correlated with increased implicit adaptation in MF participants. The Control group did not demonstrate similar significant correlations between changes in mental fatigue following watching of the documentary and explicit adaptation or implicit adaptation. Thus, it appears that increasing one's mental fatigue prior to visuomotor adaptation is associated with less engagement of strategic reaching processes and more engagement of unconscious processes.

The finding of increased implicit adaptation at the expense of explicit adaptation has been shown previously [43] and corroborated by Neville and Cressman [17], and calls into question the relationship between explicit and implicit adaptation. On one hand it has been argued that explicit and implicit adaptation are independent processes (e.g., see [12,39,40]). Alternatively, the decreased explicit adaptation and increased implicit adaptation associated with increased mental fatigue in the current experiment indicate that there may be a compensatory mechanism that looks to preserve visuomotor adaptation. For example, perhaps a reduction in engagement of explicit strategies enables implicit adaptation to occur unimpeded. At this time we put this possibility forth as a suggestion and conclude that increased mental fatigue leads to a decrease in early visuomotor adaptation, and is associated with decreased engagement of explicit adaptation.

**Retention of visuomotor adaptation.**   In addition to examining the impact of mental fatigue on visuomotor adaptation, the current research provides insight into the impact of mental fatigue prior on retention of visuomotor adaptation. As shown in Fig 4, participants demonstrated retention of visuomotor adaptation over a 20-minute rest interval.

We first looked to assess retention of explicit and implicit adaptation. Similar to our discussion of explicit and implicit adaptation above, we found no differences in retention of explicit and implicit adaptation between groups. We further found that the extent of explicit

adaptation was retained, while implicit adaptation decayed across the rest interval. Our results with respect to the retention of explicit and implicit adaptation agree with Bouchard and Cressman [44], who showed preserved explicit adaptation and limited retention of implicit adaptation following a 24-hour interval rest period.

With respect to rotated reach training trials within the retention block, we can see from Fig 4 that there is some decay of visuomotor adaptation across the rest period. That said, participants rapidly returned to how they were reaching at the end of the third reach training block, demonstrating retention of visuomotor adaptation [45]. Our more critical finding with respect to the current research question is that our MF group demonstrated a lower extent of visuomotor adaptation at the start of the retention block compared to the Control group. This decreased visuomotor adaptation was seen even though both groups had achieved the same level of visuomotor adaptation by the end of the third rotated training block and mental fatigue reports had returned to baseline levels. Moreover, both groups demonstrated similar amounts of explicit and implicit adaptation in the retention block. The continued decrease in visuomotor adaptation observed in the MF group compared to the Control group suggests a persisting impact of mental fatigue on visuomotor adaptation even when participants no longer indicated increased levels of mental fatigue. As such, an important area of study moving forward is to establish the duration for which mental fatigue impacts visuomotor adaptation by using more reach training trials and longer rest periods.

Interestingly, recent work by Neva et al. [32] employing a physically fatiguing paradigm found the oppositive impact of fatigue on visuomotor adaptation and retention in comparison to our current findings. In Neva et al.'s [32] experiment, participants engaged in an acute and intense bout of cyclic exercise for 25 minutes to induce physical fatigue prior to completing a visuomotor adaptation task. Contrary to our findings following the induction of mental fatigue, Neva et al. [32] found that fatigue arising due to physical exercise led to increased visuomotor adaptation and maintenance of visuomotor adaptation during a retention block of trials compared to control conditions. Together, our results and the findings of Neva et al. [32] call into question Kuppuswamy's [46] framework of mental fatigue, which argues that changes in effort processing, be it from mental or physical efforts, are responsible for fatigue and as such the manner by which fatigue is induced should not impact performance measures. Future research is needed to establish the relationship between mental and physical fatigue and why these two types of fatigue have been shown to differentially impact visuomotor adaptation and retention (see also [47]).

## Conclusion

Previous research has shown that visuomotor adaptation and engagement of explicit and implicit processes are influenced by the size of the visuomotor distortion introduced [16,17,39] and participants' awareness of the visuomotor distortion [16]. Our current findings establish that mental fatigue is another factor that impacts one's ability to adapt their reaches to a visuomotor distortion. Within our experiment the MF group showed reduced visuomotor adaptation in the early rotated reach training trials compared to the Control group. Given that explicit contributions have been shown to underlie early visuomotor adaptation [4,12,17], we suggest that explicit adaptation was impaired within the MF group. In support of this suggestion, we found a significant correlation between increased mental fatigue and reduced explicit adaptation in our MF group. A similar relationship was not observed in our Control group, who demonstrated a limited change in mental fatigue following the documentary watching. Finally, we found that increased mental fatigue was moderately correlated with increased implicit adaptation for our MF participants. Moving forward, the current findings indicate

that a participant's level of mental fatigue should be considered when examining visuomotor adaptation.

## Supporting information

**S1 Fig. Mental fatigue reports across the experiment.** Mental fatigue scores for the MF group (burgundy) and Control group (grey) reported across the experiment. Asterisks (*) represent significant differences relative to initial levels of mental fatigue reported at Time 1 ($p < 0.05$). Double asterisks (**) represent significant differences between groups ($p < 0.05$). (TIF)

**S2 Fig. Percent accuracy and mean reaction time across the TLDB task.** Mean performance across the 4 time intervals of the TLDB task for the Mental Fatigue group, separated by numbers (garnet) and letters (grey). **(A)** Percentage of correct responses and **(B)** mean reaction time in seconds. Error bars represent standard error of the mean. Asterisks (*) represent significant differences between consecutive time intervals in **(A)** ($p < 0.05$). Double asterisks (**) represent significant differences between stimuli in **(B)** ($p < 0.05$). (TIF)

**S1 File. Additional data analyses and results related to the Mental Fatigue Scale (MFS) and time load dual back (TLDB) task.** (DOCX)

## Author Contributions

**Conceptualization:** David Apreutesei, Erin K. Cressman.

**Data curation:** David Apreutesei.

**Formal analysis:** David Apreutesei, Erin K. Cressman.

**Funding acquisition:** Erin K. Cressman.

**Investigation:** David Apreutesei, Erin K. Cressman.

**Methodology:** David Apreutesei, Erin K. Cressman.

**Project administration:** David Apreutesei, Erin K. Cressman.

**Resources:** Erin K. Cressman.

**Software:** Erin K. Cressman.

**Supervision:** Erin K. Cressman.

**Validation:** David Apreutesei, Erin K. Cressman.

**Writing – original draft:** David Apreutesei.

**Writing – review & editing:** David Apreutesei, Erin K. Cressman.

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
