## [Decision Letter · Decision Letter 0]

16 Apr 2024

PONE-D-24-01283THE EFFECT OF MENTAL FATIGUE ON EXPLICIT AND IMPLICIT CONTRIBUTIONS TO VISUOMOTOR ADAPTATIONPLOS ONE

Dear Dr. Apreutesei,

Thank you for submitting your manuscript to PLOS ONE. After careful consideration, we feel that it has merit but does not fully meet PLOS ONE’s publication criteria as it currently stands. Therefore, we invite you to submit a revised version of the manuscript that addresses the points raised during the review process.

We look forward to receiving your revised manuscript.

Kind regards,

Karsten Witt

Academic Editor

PLOS ONE

Journal Requirements:

This work was supported by a Discovery Grant provided by the Natural Sciences and Engineering Research Council of Canada (EKC; Grant Number: RGPIN-2018-04160, url: https://www.nserc-crsng.gc.ca/professors-professeurs/grants-subs/dgigp-psigp_eng.asp). The funders had no role in this study.

3. Please ensure that you include a title page within your main document. You should list all authors and all affiliations as per our author instructions and clearly indicate the corresponding author.

Reviewers' comments:

Reviewer's Responses to Questions

**Comments to the Author**

1. Is the manuscript technically sound, and do the data support the conclusions?

Reviewer #1: Partly

Reviewer #2: Partly

2. Has the statistical analysis been performed appropriately and rigorously? 

Reviewer #1: Yes

Reviewer #2: I Don't Know

3. Have the authors made all data underlying the findings in their manuscript fully available?

Reviewer #1: Yes

Reviewer #2: No

4. Is the manuscript presented in an intelligible fashion and written in standard English?

Reviewer #1: Yes

Reviewer #2: Yes

5. Review Comments to the Author

**Reviewer #1:** The study is of great interest and well conducted.

However, in my opinion, there are some major issues that need to be addressed in the paper.

First of all, too much emphasis is placed on the protocol used to induce Mental Fatigue. While reading the paper, this aspect shifts the focus, creating confusion and ambiguity. Induction of Mental Fatigue is an important aspect but aimed to create the conditions for studying its effects on motor adaptation.

I would therefore ask the authors:

1) to re-formulate the abstract: it is mainly focused on the techniques for inducing mental fatigue and describes too briefly and summarily the results obtained on motor adaptation;

2) to re-formulate the introduction. The objectives of the study must be well formulated for understanding how mental fatigue influences motor adaptation. I consider that it is necessary for motor adaptation to be well described in all its components: learning, adaptation, implicit and explicit adaptation. Next, the concept of mental fatigue must be introduced. At this point, explain the importance of studying if and how mental fatigue affects motor adaptation, because of the implications of this influence in everyday life. Finally, I suggest formulating the experimental objectives clearly and concisely.

3) the conclusions must be also reviewed according to a focused outline, enabling the reader to follow a clear and linear line of thought.

4) in the results section, p. 26, please, remove the sentence "mental fatigue reduced participants' ability to adapt their reaches to the visuomotor distortion": mental fatigue slows down the learning process but does not reduce this ability.

5) in the conclusions section, please remove the sentence "Here we show that mental fatigue reduces visuomotor adaptation and retention due to decreased engagement of explicit processes." De facto, colleagues find only a correlation between fatigue levels and adaptation times, in fatigued subjects, but not a difference in the performance of these compared to the performance of the control group were found. This correlation may suggest a possible causal relationship, but, in this study, it is not possible to draw these conclusions, since data do not support them.

**Reviewer #2:** The study considers the effect of mental fatigue on visuomotor adaptation. Specifically explicit and implicit processes, which are defined as conscious (explicit) and unconscious (implicit) strategies in a reaching task. Mental fatigue was experimentally triggered using a time load dual back (TLDB) task and compared to a control group that was watching a documentary. After that, both groups completed several reaching tasks and rated their subjective mental fatigue. The hypotheses were that mental fatigue has an impact on visuomotor adaptation and therefore the experimental group would take longer to adapt their movements in the reaching task. Further, mental fatigue was expected to impact explicit contributions more than implicit contributions. The results show that the experimental group did take longer to adapt to the reaching task and that the mental fatigue did not lead to a decrease in the use of conscious strategies. However there was an increase of the use of unconscious strategies between the first two reaching blocks, which was first stable and then decreasing again after a 20 minute resting period.

MAJOR CHANGES

1. I have concerns, whether the assumptions for an ANOVA analysis/ t-test are always met. If all assumptions are checked, the authors should add this in a short sentence or give evidence in the appendix. If this is not the case, e.g. non-parametric tests and Linear Mixed Model need to be used for the analysis.

2. The background of some of the statistical analysis is not comprehensible. The authors should clarify the underlying question for the rmANOVA analysis for the TLDB and MFS as well as the correlation analysis. To justify the analysis there needs to be a research question and hypothesis for these analyses.

3. The conclusion of this study is that mental fatigue leads to a “decreased engagement of explicit processes”, however in the result section no significant effect has been observed for explicit processes. Instead, the effect on implicit processes is not mentioned in the conclusion. The authors should clarify the conclusion section to avoid confusion.

4. The interpretation of the correlation analysis should be re-evaluated. A correlation coefficient is an effect size measure. When talking about the effect, the authors should state the direction and strength of the effects, rather than its significance. Also a correlation does not prove a causal effect.

5. The structure of the manuscript does not appear to be sound.

For example the paragraph

„Two groups of participants (Mental Fatigue (MF) and Control groups) initially reached with aligned cursor feedback. Following this, mental fatigue was induced in the MF group of participants through the TLDB task of Borraganet al. (2016). All participants then reached with rotated cursor feedback in an adaptation phase for 45 trials, such that the cursor motion was rotated 40° CW relative to hand motion. Explicit and implicit adaptation were assessed following rotated reach training using the PDP (Modchalingham et al., 2019; Neville and Cressman, 2018; Werner et al., 2015). Participants completed the rotated reach training trials and PDP trials 3 more times to track explicit and implicit contributions to visuomotor adaptation over time, with the final block administered following a 20-minute rest in order to assess retention. Throughout the experiment, mental fatigue was assessed by having participants complete the MFS adapted from Lee and colleagues’ (1991) VAS-F.“

is already explaining the experimental setting in detail. In this length it does not reflect the scope of the study and should rather be included in the method section.

An example for the Results section is e.g for the last paragraph of „late visoumotor adaption“ and the last paragraph of „implicit adaptation“. An interpretation of the results does not belong in the results sections but needs to be stated in the discussion section.

Further, in the discussion the TLDB task does not need to be explained again as it can be assumed to be known from the method section.

These are examples, the authors should thoroughly re-evaluate the manuscript to correctly place the information in the formally right sections and to avoid redundance.

6. It would be beneficial to report the mean age of the two groups. Since the age range is from 18-40 and the overall mean age is 19.8 (+/- 1.7), there is a potential for a significant age difference between groups which can be a bias to the effects.

7. Figure 4a: is the decrease for the MF group between timepoint 3 and 4 significant? The discussion is missing why there may be a decreasing trend between that two timepoints and an increasing trend after that.

MINOR CHANGES

1. The Matlab Program that was used to analyse the reaching trials needs to be explained in a bit more detail or accessible in the appendix

There are major points that need to be considered in this study to provide a comprehensable structure.

6. PLOS authors have the option to publish the peer review history of their article (what does this mean?). If published, this will include your full peer review and any attached files.

Reviewer #1: No

Reviewer #2: No

---

## [Author Response · Author response to Decision Letter 0]

31 May 2024

Dear Dr. Witt,

Thank you to the Reviewers and yourself for taking the time to review our manuscript entitled “The effect of mental fatigue on explicit and implicit contributions to visuomotor adaptation” (PONE-D-24-01283). Below we outline how we have revised the manuscript to address the comments raised by the Reviewers and yourself (numbered and in bold). In addition, our revised manuscript has been uploaded with tracked changes (red text), which take into account the questions and suggestions raised by the Reviewers. 

Thank you again for providing such helpful feedback. 

David Apreutesei, MSc

Erin K. Cressman, PhD

Academic Editor

The manuscript and figure files have been reformatted to meet PLOS ONE’s style and naming requirements.

2. Please provide an amended statement that declares all the funding or sources of support (whether external or internal to your organization) received during this study, as detailed online in our guide for authors at http://journals.plos.org/plosone/s/submit-now. Please also include the statement “There was no additional external funding received for this study.” in your updated Funding Statement. Please include your amended Funding Statement within your cover letter. We will change the online submission form on your behalf.

We have amended our funding statement to read: 

This work was supported by a Discovery Grant provided by the Natural Sciences and Engineering Research Council of Canada (EKC; Grant Number: RGPIN-2018-04160). The funders had no role in this study. There was no additional external funding received for this study. 

3. Please ensure that you include a title page within your main document. You should list all authors and all affiliations as per our author instructions and clearly indicate the corresponding author.

A title page has been included with our main document. Specifically, we have included all authors and affiliations and clearly indicate the corresponding author.

We have amended our list of authors so that each author is linked to an institute.

We have updated our ethics statement in the paper to read (Lines 107-109): 

This experiment was approved by the University of Ottawa Health Sciences and Science Research Ethics Board. All participants provided written informed consent prior to testing and were informed that they could withdraw at any time.

As indicated in our updated ethics statement, the full name of our institution’s research ethics board is the “University of Ottawa Health Sciences and Science Research Ethics Board”.

Reviewer 1 

The study is of great interest and well conducted. However, in my opinion, there are some major issues that need to be addressed in the paper. First of all, too much emphasis is placed on the protocol used to induce Mental Fatigue. While reading the paper, this aspect shifts the focus, creating confusion and ambiguity. Induction of Mental Fatigue is an important aspect but aimed to create the conditions for studying its effects on motor adaptation. I would therefore ask the authors:

1. To re-formulate the abstract: it is mainly focused on the techniques for inducing mental fatigue and describes too briefly and summarily the results obtained on motor adaptation.

Thank you for this suggestion. We now have removed much of our discussion of the time-load dual back task used to induce mental fatigue from the abstract and additional results regarding visuomotor adaptation have been included (see lines 24-32). 

2. To re-formulate the introduction. The objectives of the study must be well formulated for understanding how mental fatigue influences motor adaptation. I consider that it is necessary for motor adaptation to be well described in all its components: learning, adaptation, implicit and explicit adaptation. Next, the concept of mental fatigue must be introduced. At this point, explain the importance of studying if and how mental fatigue affects motor adaptation, because of the implications of this influence in everyday life. Finally, I suggest formulating the experimental objectives clearly and concisely.

In the current research we ask if mental fatigue influences motor adaptation, specifically explicit and implicit adaptation. Thus, as the reviewer suggests, it is critical that the reader understand all components of motor adaptation (including adaptation, as well as implicit and explicit adaptation). We have thus reformatted our entire introduction as suggested by the reviewer, first introducing visuomotor adaptation, and then explaining why we are looking at the influence of cognitive fatigue on visuomotor adaptation and the implications of this in everyday life. The experimental objectives and hypotheses have been reworded to be clear and concise, such that at Lines 87-97 we now state:

In the current study we asked if mental fatigue impacts visuomotor adaptation and underlying explicit and implicit contributions over time. Participants trained to reach with a large cursor rotation (i.e., 40°), that has consistently been shown to engage both explicit and implicit processes [16, 17]. Given that mental fatigue has been shown to impact cognitive performance and motor control (see [19, 22, 31]), it was hypothesized that the MF group would show decreased visuomotor adaptation compared to the Control group early in training, when explicit processes have been shown to be engaged [4, 17]. Furthermore, it was hypothesized that explicit visuomotor adaptation as established via the PDP would be less in the MF group compared to the Control group over the course of the experiment. These results would indicate that mental fatigue impacts the conscious strategic component of visuomotor adaptation, while leaving the unconscious contribution largely unaffected. 

3. The conclusions must be also reviewed according to a focused outline, enabling the reader to follow a clear and linear line of thought.

We have revised our Discussion and Conclusion, ensuring that the results related to our objectives are stated at the beginning of the Discussion (see Lines 474-477). Moreover, our conclusion now reads as followed on Lines 661-675: 

 Previous research has shown that visuomotor adaptation and engagement of explicit and implicit processes are influenced by the size of the visuomotor distortion introduced [16, 17, 39] and participants’ awareness of the visuomotor distortion [16]. Our current findings establish that mental fatigue is another factor that impacts one’s ability to adapt their reaches to a visuomotor distortion. Within our experiment the MF group showed reduced visuomotor adaptation in the early rotated reach training trials compared to the Control group. Given that explicit contributions have been shown to underlie early visuomotor adaptation [4, 12, 17], we suggest that explicit adaptation was impaired within the MF group. In support of this suggestion, we found a significant correlation between increased mental fatigue and reduced explicit adaptation in our MF group. A similar relationship was not observed in our Control group, who demonstrated a limited change in mental fatigue following the documentary watching. Finally, we found that increased mental fatigue was moderately correlated with increased implicit adaptation for our MF participants. Moving forward, the current findings indicate that a participant’s level of mental fatigue should be considered when examining visuomotor adaptation.

4. In the results section, p. 26, please, remove the sentence "mental fatigue reduced participants' ability to adapt their reaches to the visuomotor distortion": mental fatigue slows down the learning process but does not reduce this ability.

Thank you for drawing our attention to this statement. We have removed this statement and now make sure to clarify that deficits are observed with respect to initial visuomotor adaptation (see Lines 403-412):

Visuomotor adaptation differed between Groups during initial rotated reach training trials across the 4 blocks, as indicated by a significant main effect of Group (F(1,38) = 4.817, p = 0.034). Overall, participants in the MF group demonstrated less initial visuomotor adaptation than participants in the Control group across all 4 blocks of rotated training (MF: X̅ = 25.56°, SE = 0.76°; Control: X̅ = 28.58°, SE = 1.15°). Analyses also revealed a significant main effect of Block (F(2.3, 85.5) = 128.535, p < 0.001), but no significant Block x Group interaction (F(2.3, 85.5) = 0.885, p = 0.427). Post-hoc analysis confirmed that early visuomotor adaptation increased across the first three Blocks, with PVAE significantly greater in Block 2 compared to Block 1 (p < 0.001) and Block 3 compared to Block 2 (p < 0.001). There was no difference in early PVAE between Block 3 and Block 4, following the 20-minute rest (p = 1). 

5. In the conclusions section, please remove the sentence "Here we show that mental fatigue reduces visuomotor adaptation and retention due to decreased engagement of explicit processes." De facto, colleagues find only a correlation between fatigue levels and adaptation times, in fatigued subjects, but not a difference in the performance of these compared to the performance of the control group were found. This correlation may suggest a possible causal relationship, but, in this study, it is not possible to draw these conclusions, since data do not support them.

The reviewer is correct. We do not have any evidence directly linking the reduction observed in visuomotor adaptation as being due to decreased engagement of explicit processes. We do show that early visuomotor adaptation is less in our MF group vs. Control group, where early visuomotor adaptation has shown to engage explicit processes (see Taylor et al., 2014; Neville and Cressman, 20198). As well, we do have a negative moderate correlation between mental fatigue and explicit adaptation for participants in our MF group. 

We have reworded our conclusions to reflect that results observed and put forth a suggestion (Lines 669-675):

Within our experiment the MF group showed reduced visuomotor adaptation in the early rotated reach training trials compared to the Control group. Given that explicit contributions have been shown to underlie early visuomotor adaptation [4, 12, 17], we suggest that explicit adaptation was impaired within the MF group. In support of this suggestion, we found a significant correlation between increased mental fatigue and reduced explicit adaptation in our MF group. A similar relationship was not observed in our Control group, who demonstrated a limited change in mental fatigue following the documentary watching.

Reviewer 2

The study considers the effect of mental fatigue on visuomotor adaptation. Specifically explicit and implicit processes, which are defined as conscious (explicit) and unconscious (implicit) strategies in a reaching task. Mental fatigue was experimentally triggered using a time load dual back (TLDB) task and compared to a control group that was watching a documentary. After that, both groups completed several reaching tasks and rated their subjective mental fatigue. The hypotheses were that mental fatigue has an impact on visuomotor adaptation and therefore the experimental group would take longer to adapt their movements in the reaching task. Further, mental fatigue was expected to impact explicit contributions more than implicit contributions. The results show that the experimental group did take longer to adapt to the reaching task and that the mental fatigue did not lead to a decrease in the use of conscious strategies. However there was an increase of the use of unconscious strategies between the first two reaching blocks, which was first stable and then decreasing again after a 20 minute resting period.

MAJOR CHANGES

1. I have concerns, whether the assumptions for an ANOVA analysis/ t-test are always met. If all assumptions are checked, the authors should add this in a short sentence or give evidence in the appendix. If this is not the case, e.g. non-parametric tests and Linear Mixed Model need to be used for the analysis.

Thank you for this comment. We have now noted that statistical assumptions were met or non-parametric tests used (see Lines 243-251):

Following confirmation that data met the required assumptions for the statistical test used, statistical analyses were completed using parametric statistics in IBM SPSS Statistics for Windows version 28.0 unless otherwise noted. The Greenhouse-Geisser correction was applied when appropriate and corresponding p-values are reported. For data that were not normally distributed according to the Kolmogorov-Smirnov test (see Final adaptation: Late training), the non-parametric Mann-Whitney U test was used to test for group differences. The significance value for all statistical tests was set at p < 0.05 and Bonferonni post-hoc tests corrected for multiple comparisons were used to find the locus of significant effects or interactions when required.

2. The background of some of the statistical analysis is not comprehensible. The authors should clarify the underlying question for the rmANOVA analysis for the TLDB and MFS as well as the correlation analysis. To justify the analysis there needs to be a research question and hypothesis for these analyses.

Thank you. This is a good point to raise. We have now justified all the analyses included in the manuscript (e.g., analyses related to the TLDB task and MFS, as well as correlation analyses). For the TLDB task, we wanted to ensure that all participants continued to engage in the task over the duration of the 20 minutes (i.e., demonstrated no declines in performance). In the manuscript we thus compared percentage of correct responses and reaction time for initial (i.e., first 25%) and final (i.e., last 25%) trials completed. Additional analyses related to stimulus type and all four time intervals can be found in the Supplementary File. 

With respect to the TLDB task we now only include results related to mental fatigue pre and post TLDB task or documentary watching, in order to determine if our mental fatigue intervention worked. Additional analyses comparing mental fatigue over the entire duration of the experiment and between groups at additional time periods is provided in the Supplementary File. 

Finally, with respect to the correlational analyses, additional details re justification are provided on Lines 346-350:

Finally, given the variance we observed across participants in the MF group with respect to changes in level of mental fatigue following the TLDB task and our hypothesis that increased mental fatigue would be associated with decreased explicit adaptation, one-tailed Pearson correlation analyses were completed to assess the relationship between both explicit and implicit adaptation and mental fatigue for the MF and Control groups. 

Also to note, we have taken out factors in some of our ANOVAs. For example, we no longer include the factor of Bin when analyzing early visuomotor adaptation, as we did not have a hypothesis for this factor. 

3. The conclusion of this study is that mental fatigue leads to a “decreased engagement of explicit processes”, however in the result section no significant effect has been observed for explicit processes. Instead, the effect on implicit processes is not mentioned in the conclusion. The authors should clarify the conclusion section to avoid confusion.

Thank you. This was also pointed out 

---

## [Editor Report · Decision Letter 1]

11 Jul 2024

The effects of mental fatigue on explicit and implicit contributions to visuomotor adaptation

PONE-D-24-01283R1

Dear Dr. Cressman,

We’re pleased to inform you that your manuscript has been judged scientifically suitable for publication and will be formally accepted for publication once it meets all outstanding technical requirements.

Kind regards,

Karsten Witt

Academic Editor

PLOS ONE
---

## [Editor Report · Acceptance letter]

8 Aug 2024

PONE-D-24-01283R1 

PLOS ONE

Dear Dr. Cressman, 

I'm pleased to inform you that your manuscript has been deemed suitable for publication in PLOS ONE. Congratulations! Your manuscript is now being handed over to our production team.

Kind regards, 

on behalf of

Dr. Karsten Witt 

Academic Editor

PLOS ONE